# OpenReview forum: "Reducing Hallucinations in Multimodal Large Language Models via Causal Fusion"
_ICLR.cc/2026/Conference — ICLR 2026 Conference Withdrawn Submission_

### Official Review · Reviewer_oNr2 · 2025-10-28

**Soundness:** 3
**Presentation:** 3
**Contribution:** 2
**Rating:** 4
**Confidence:** 3

**Summary:**

This paper proposes COAD, which aims  to mitigate object hallucination in VLM by incorporating causal inference into the decoding process. It first uses an external object detector to obtain object probabilities from the image. An MLLM is then finetuned to condition its output on these object probabilities, alongside the image and previous text. During inference, COAD estimates an oracle next-token prediction by performing a causal intervention and fusing the outputs of the original pretrained model and the finetuned model based on a derived causal formula. Experiments primarily on LLaVA-1.5-7B show that COAD reduces hallucination rates on benchmarks like CHAIR, MMHal-Bench, and POPE compared to several baseline decoding strategies.

**Strengths:**

1.The paper introduces a causal perspective to analyze and mitigate object hallucination, attempting to disentangle the influence of visual input from potentially spurious correlations introduced by previously generated text.

2.COAD demonstrates improved performance in reducing object hallucination rates compared to several existing decoding-based methods on benchmarks like CHAIR, MMHal-Bench, and POPE, particularly when applied to the LLaVA-1.5-7B model.

**Weaknesses:**

1.The proposed method appears overly complex, requiring multiple components and stages. It necessitates an external object detector to generate object proposals, a separate finetuning stage for the MLLM to incorporate these proposals, and a final dual-MLLM contrastive decoding step that combines logits from both pretrained model and finetuned model during inference. This multi-step process adds significant complexity compared to simpler decoding strategies.

2.A major concern is the choice of the base model used for evaluation, LLaVA-1.5-7B, which is significantly outdated compared to current sota VLMs like Qwen2.5VL or InternVL3. There is a substantial performance gap between LLaVA-1.5 and modern VLMs, raising doubts about whether the causal mechanisms and improvements observed with COAD would still be relevant or effective on these more capable and potentially inherently less hallucinatory models. The lack of validation on sota models severely limits the assessment of COAD's practical impact.

3.Regarding the CHAIR evaluation, clarification on the specific setup would be appreciated. The reported CHAIR scores for LLaVA-1.5 7B and OPERA appear to differ somewhat from those presented in the OPERA paper. Could the authors elaborate on the evaluation details to help reconcile these observations?

**Questions:**

Please refer to the weakness.

---

### Official Review · Reviewer_CiZz · 2025-10-29

**Soundness:** 1
**Presentation:** 2
**Contribution:** 1
**Rating:** 2
**Confidence:** 5

**Summary:**

This paper introduces a decoding method aimed at reducing hallucinations. At a high level, the proposed approach works as follows: first, ground-truth objects are extracted from an image using a separate object detection model, serving as an oracle. During inference, causal decoding is intervened whenever a generated object is not present in the precomputed oracle. The method is evaluated on the MLLM LLaVA 1.5 across three object hallucination benchmarks.

**Strengths:**

- The idea of intervening in the causal decoding to rule out spurious correlations that may lead to object hallucination is quite interesting.
- This paper addresses object hallucination, which remains a critical problem in MLLMs. And, spurious correlations indeed cause hallucinations in MLLMs.

**Weaknesses:**

* The experimental setup is not clearly described. What is the difference between the pre-trained and fine-tuned MLLM in this context? Could you make this explicit? Does the pre-trained MLLM refer to the checkpoint before SFT? In most practical scenarios, we only have access to the final checkpoint—how would your method be adapted in such cases?

* Some low-level details of the proposed method are unclear. Could you provide a complete example with a sample image and question to illustrate how the process works? A simple pseudo-code would also help clarify the pipeline. Are you planning to release the code? You could even share an anonymous repository during review.

* A key limitation of this approach is that it becomes ineffective when the model response does not explicitly mention the object name. For instance, when describing an image containing a cat, the MLLM will generate the word “cat” and your method can calculate the probability of cat present in the precomputed objects. However, if the question is “Is there a cat?” and the answer is simply “Yes,” your inference formulation seems inapplicable. This likely explains the weaker results on POPE. Furthermore, how does your method generalize to questions not about object existence, but about object attributes or relations? In such cases, the pre-identification of objects seems to be not meaningful.

* The experimental setup is rather limited, as it evaluates only on a single MLLM (LLaVA 1.5), which is relatively outdated (released in 2023). The three chosen benchmarks are also not particularly robust. POPE assesses object existence based on just 500 images and does not cover other hallucination types such as attributes or relations. Similarly, CHAIR uses 500 MSCOCO images with limited ground-truth annotations, and MMHal-Bench contains only 96 images. I suggest evaluating on more comprehensive benchmarks such as HallusionBench, AMBER, M-HalDetect, and GAVIE.

* The comparison setup is also weak. The authors are encouraged to include comparisons with both offline and online RL-based hallucination mitigation methods.

**Questions:**

Please see weaknesses.

---

### Official Review · Reviewer_Q6LA · 2025-11-01

**Soundness:** 2
**Presentation:** 3
**Contribution:** 2
**Rating:** 4
**Confidence:** 4

**Summary:**

This paper proposes a causal-based approach to reducing hallucinations. The authors argue that certain hallucinations arise not solely from the input image itself, but from prior hallucinated text that subsequently triggers further hallucinations. Based on this observation, the authors introduce a mechanism to detect whether an output token has a causal relationship with the input image, and leverage this mechanism to mitigate hallucinations in multimodal large language models.

**Strengths:**

The proposed method is novel, representing the first attempt to apply causal inference to mitigate hallucinations.

**Weaknesses:**

1. The proposed method relies on an external object detection tool, which itself is trained on large-scale datasets. Therefore, the proposed approach should also be categorized as one that leverages external knowledge, rather than, as the authors claim, relying solely on internal mechanisms. Consequently, comparing this method with those designed for "internal hallucination mitigation" is not entirely fair.

2. The experiments do not report the length of the output tokens. The length of the generated output is crucial for assessing hallucination, as longer outputs generally have a higher risk of hallucination. Mathematically speaking, in an extreme case, a model that outputs zero tokens would exhibit no hallucination at all.

3. On the MSCOCO dataset, the proposed method does not show a significant improvement compared with previous approaches.

4. The paper lacks experiments validating the effectiveness of the proposed causal mechanism itself. How can we be sure that the observed performance gains are due to the causal model’s capability rather than other factors? It is also possible that, in some cases, a poorly performing causal model might actually lead to fewer hallucinations.

**Questions:**

see Weaknesses

---

### Official Review · Reviewer_vfM4 · 2025-11-01

**Soundness:** 2
**Presentation:** 2
**Contribution:** 1
**Rating:** 2
**Confidence:** 4

**Summary:**

The paper proposes Causal Object-Aware Decoding (COAD), a new framework to reduce hallucinations in Multimodal LLMs. COAD involves finetuning the MLLM (LLaVA) to accept an additional input vector derived from an external object detector that processes the image. At inference, COAD uses a causal formulation to combine the logit outputs of this new finetuned model with the original model. The authors show this approach causally intervenes in the generation process, leading to state-of-the-art faithfulness by grounding the output in the detected objects.

**Strengths:**

- Reducing object hallucination in MLLMs is a timely and high-impact problem for the reliability of generative AI systems.
- The idea of leveraging external information, like an object detector's output, to ground and aid the generation is an interesting and sensible direction.
- The paper is clearly written.

**Weaknesses:**

- COAD is a finetuning-based method that requires architectural modifications and a new training phase. The paper compares it against inference-only, training-free decoding strategies. Given that COAD has seen additional supervised signals and has been optimized specifically for this task, the comparison is unfair. Other finetuning or alignment methods for hallucination reduction, e.g., RLHF or DPO-based, which do not require architectural modifications,  are entirely missing from the baselines.
- The method is heavy. At every decoding step, it requires running the original model, running the finetuned MLLM, and running the external object detector (sampled several times). This is computationally prohibitive and makes the method impractical compared to the lightweight, inference-only baselines or even alignment methods that do not add computational overhead during inference.
- COAD relies on a closed-vocabulary object detector. In my opinion, this is a major limitation and a conceptual step backward. Multimodal LLMs are powerful because of their open-vocabulary nature. This method can only de-hallucinate objects within the detector's pre-defined class list, and it's unclear how it would handle – or if it would worsen – hallucinations of objects not in that list. In fact, requiring architectural modifications, adding new objects would likely require restarting from scratch.
- COAD is only validated on a single model architecture at fixed, small size (LLaVA v1.5 7B). Varying model scale and benchmarking it with at least another architecture are necessary to claim general causal frameworks for multimodal LLMs.
- The paper only reports on correctness (CHAIR), not coverage. Coverage-related metrics, such as the fraction of present objects that are mentioned, are not reported. This is a critical omission, as a model could achieve a perfect CHAIR score by simply not mentioning any objects (e.g., "This is a nice picture."), which is a useless output. There is a fundamental trade-off between correctness and coverage that is not discussed.
- Table 4 suggests that the paper's primary contribution is not the "causal fusion" but rather the simple act of finetuning with an object detector.
- In the first example in Appendix E, the COAD caption is completely incorrect. The base LLaVA model's caption, while hallucinating a bench, is arguably more faithful to the overall scene.
- The theoretical model (involving an oracle) – and the scope – is quite close to that of Favero et al., 2024 (Multi-Modal Hallucination Control by Visual Information Grounding), which is relevant and uncited.

**Questions:**

- How does COAD handle objects not in the detector's vocabulary list?
- Could you provide a detailed analysis of the inference cost for COAD vs. baselines?

---

### Note · Authors · 2025-11-20

I have read and agree with the venue's withdrawal policy on behalf of myself and my co-authors.